# S²-CSNet: Scale-Aware Scalable Sampling Network for Image Compressive Sensing

Chen Hui
Harbin Institute of Technology
Harbin, China
chenhui@stu.hit.edu.cn

Haiqi Zhu
Harbin Institute of Technology
Harbin, China

Shuya Yan
Harbin Engineering University
Harbin, China

Shaohui Liu*
Harbin Institute of Technology
Harbin, China

Feng Jiang*
Harbin Institute of Technology
Harbin, China

Debin Zhao
Harbin Institute of Technology
Harbin, China

## Abstract

Deep network-based image Compressive Sensing (CS) has attracted much attention in recent years. However, there still exist the following two issues: 1) Existing methods typically use fixed-scale sampling, which leads to limited insights into the image content. 2) Most pre-trained models can only handle fixed sampling rates and fixed block scales, which restricts the scalability of the model. In this paper, we propose a novel scale-aware scalable CS network (dubbed S²-CSNet), which achieves scale-aware adaptive sampling, fine granular scalability and high-quality reconstruction with one single model. Specifically, to enhance the scalability of the model, a structural sampling matrix with a predefined order is first designed, which is a universal sampling matrix that can sample multi-scale image blocks with arbitrary sampling rates. Then, based on the universal sampling matrix, a distortion-guided scale-aware scheme is presented to achieve scale-variable adaptive sampling, which predicts the reconstruction distortion at different sampling scales from the measurements and select the optimal division scale for sampling. Furthermore, a multi-scale hierarchical sub-network under a well-defined compact framework is put forward to reconstruct the image. In the multi-scale feature domain of the sub-network, a dual spatial attention is developed to explore the local and global affinities between dense feature representations for deep fusion. Extensive experiments manifest that the proposed S²-CSNet outperforms existing state-of-the-art CS methods.

## CCS Concepts

• **Computing methodologies** → **Modeling and simulation**.

## Keywords

Image compressive sensing, scale-aware sampling, model scalability, image compression, deep networks, image restoration

*Corresponding authors.

**ACM Reference Format:**
Chen Hui, Haiqi Zhu, Shuya Yan, Shaohui Liu, Feng Jiang, and Debin Zhao. 2024. S²-CSNet: Scale-Aware Scalable Sampling Network for Image Compressive Sensing. In *Proceedings of the 32nd ACM International Conference on Multimedia (MM '24), October 28-November 1, 2024, Melbourne, VIC, Australia.* ACM, New York, NY, USA, 10 pages. https://doi.org/10.1145/3664647.3681323

## 1 Introduction

Compressive sensing (CS) is an effective signal processing technique, which has generated significant research interest in the signal and image processing communities [3, 77]. Mathematically, supposing that $x \in \mathbb{R}^{n \times 1}$ is an input signal, CS can achieve fast imaging by sampling far fewer measurements than that required by Nyquist sampling [10, 14, 31], i.e., $y = \Phi x$, where $y \in \mathbb{R}^{m \times 1}$ are the observed measurements, $\Phi \in \mathbb{R}^{m \times n}$ with $m \ll n$ is the sampling matrix and $\frac{m}{n}$ is defined as the sampling rate (or CS ratio). Due to the simple and fast sampling, CS has been widely deployed including snapshot compressive imaging [40, 71], medical imaging [54, 56], image encryption [35], and compressive learning [50, 66, 87].

In the development of image CS, block-based CS (BCS) [6, 19, 30] has emerged as a classical method and has been widely adopted by most research efforts [5, 29, 58, 80, 88]. In BCS, images are divided into non-overlapping blocks of fixed scale and sampled block by block. Following BCS, some representative sampling matrices have been proposed, including the structural matrix [8, 20], the random matrix [63, 68] and the binary matrix [2, 44]. Corresponding to these sampling methods, some model-based reconstruction methods [13, 21, 86] have been presented, which usually utilize various iterative solvers to reconstruct images. However, due to extensive fine-tuning, these methods incur high computational costs.

Driven by the powerful learning ability of deep neural networks, deep network-based CS has demonstrated superior performance compared to traditional sampling and reconstruction methods. In these works, they explore the use of fully connected layers [1, 42] and the convolutional layers [12, 39, 58] to model the sampling matrix, and propose network-based deep reconstruction methods. Some earlier studies adopt block-by-block reconstruction [34, 51, 72, 78], which ignores the correlation between different blocks in the image, leading to serious blocking artifacts. To address this issue, some recent works [15, 58, 60, 84] feed all measurements of all blocks jointly into the reconstruction network and effectively eliminate the blocking artifacts. However, one of the weaknesses of these deep CS networks is that the sampling process ignores the

image content and applies a spatially uniform sampling rate to the entire image. Considering that the meaningful information in an image is usually not uniformly distributed, some researchers have proposed adaptive CS [5, 53, 80, 88], which can adaptively allocate the sampling rate according to the content features of the image.

By treating each block independently and dynamically allocating the sampling rate, adaptive CS achieves content-aware sampling and provides further intrinsic insight into the image content. However, the existing adaptive CS methods suffer from the following issues: 1) Model scalability: Following [5, 57], we define fine granular scalability as a model that can handle arbitrary CS ratios and allocate the CS ratio adaptively according to the image content. By mapping image content features to sampling rates, some adaptive CS [53, 88] employ a multi-channel solution with each channel handling a single sampling rate, which limits the scalability of the model. 2) Image prior computation: Some ideal solutions [80, 88] directly use the original image to calculate prior features for guiding the adaptive sampling, which is not always possible to access the complete image before CS sampling [5, 29]. 3) Block-based sampling: Existing adaptive CS schemes mechanically divide the image by a fixed scale, such coarse blocking leads to the mixing of information with different content features (e.g., texture regions and smooth regions), which will cause uneven sampling rate allocation and potentially degrade the performance of adaptive CS.

To overcome above issues, we propose a novel scale-aware scalable CS network ($S^2$-CSNet). Specifically, to enhance the scalability of the model, a structured learnable matrix with a predefined order is first designed, which can sample multi-scale image blocks with arbitrary sampling rates. Then, to address the issue of image prior computation, a distortion estimation method based on the universal sampling matrix is introduced, which estimates the real reconstruction distortion in the measurement domain. Then, a scale-aware sampling scheme is presented to achieve multi-scale sampling, which calculates the reconstruction distortion under different scale partitions of the same region and selects the partition mode with the minimum distortion for sampling. Finally, a multi-scale hierarchical sub-network under a well-defined compact framework is put forward to efficiently reconstruct the image. In the multi-scale feature domain, a dual spatial attention mechanism is suggested to explore the local and global affinities between dense feature representations for deep feature fusion.

The main contributions are summarized as follows:

**(1)** We propose a novel scale-aware scalable CS network $S^2$-CSNet, which adopts a different approach from the traditional BCS and achieves scale-aware adaptive sampling and fine granular scalability without direct access to the original image.

**(2)** In $S^2$-CSNet, to achieve scale-aware sampling, a universal sampling matrix is designed, which can sample multi-scale image blocks with arbitrary sampling rates. Based on this sampling matrix, a distortion-guided scale-aware scheme is presented, which can evaluate the reconstruction distortion under multi-scale partitioning and select the optimal sampling scale.

**(3)** A multi-scale hierarchical sub-network under a well-defined compact framework is put forward to reconstruct the image, in which a dual spatial attention mechanism is developed to explore the local and global affinities between dense feature representations to further enhance the reconstruction ability.

## 2 Related Work

### 2.1 Block-based Image CS Sampling

As a lightweight sampling method, BCS effectively reduces computational complexity and is widely used in image CS [37, 80, 88]. In image BCS, the sampling process can be described as $y_i = \Phi_{s,B} x_i^B$, where $x_i^B$ denotes the $i$-th image block with spatial size $B$ and channel number $l$, $\Phi_{s,B} \in \mathbb{R}^{m \times lB^2}$ with $m = \lfloor s \times lB^2 \rfloor$ is a predefined sampling matrix for sampling rate $s$. Based on the above idea, some researchers propose deep network-based CS methods [1, 5, 12, 29, 42, 58], which use convolutional layers to represent the sampling matrix and achieve remarkable results. However, these methods are content-independent and cannot adaptively allocate the sampling rate according to the image content features.

In adaptive CS, the sampling rate $s_i \in \{m/Q\}_{m=1}^Q$ with $Q = lB^2$ of an image block can be an arbitrary value between 0 and 1 [5, 29]. If non-adaptive CS is applied to solve this problem, e.g., [58, 76, 84, 88], they need to train $Q$ models to satisfy arbitrary sampling rates. To sample images at different sampling rates in a single model, some works propose to train multiple sampling matrices [62, 73, 76]. However, such an operation will impose a large storage burden, with a total memory cost of $\sum_{m=1}^Q (mQ) = [Q^2(Q+1)/2] \in O(Q^3)$ [5, 29] if $Q$ sampling matrices need to be trained. To address this problem, some researchers divide the sampling matrix into different hierarchies to handle different sampling rates [43, 57, 83]. Based on these works, some recent CS methods [5, 29, 53, 80, 88] implement adaptive sampling rate allocation. In particular, [5] and [29] employ an auxiliary lightweight branch to compute prior features from measurements and train a universal sampling matrix to handle arbitrary sampling rates. Although these methods achieve adaptive sampling rate allocation, they all mechanically use fixed-scale blocking, which affects the performance of adaptive sampling.

### 2.2 Image CS Reconstruction

Recovering the original signal $x$ from a extremely small number of measurements $y$ is a classical ill-posed problem. The traditional CS method solves this problem in an optimized manner:

$$\hat{x} = \arg\min_x \frac{1}{2} ||\Phi x - y||_2^2 + \delta\lambda(x) \tag{1}$$

where $\delta\lambda(x)$ is a prior term with regularization parameter $\delta$. Based on Eq. (1), many effective methods have been proposed, such as gradient descent methods [16], greedy methods [45, 67], and convex optimization methods [70]. For image CS, more complex structures are designed, including prior-based methods [85], projected Landweber-based methods [18], and denoiser-based methods [55, 82]. However, these traditional methods require constant iteration, leading to a high computational complexity.

Different from traditional CS methods, deep network-based CS exhibits greater potential. Specifically, early works [34, 74] often rely on block-based CS, where the target image is reconstructed block by block, and then all reconstructed image blocks are concatenated together to form the final image. However, connecting all the blocks in such a coarse manner will lead to serious blocking artifacts, especially at low sampling rates [27, 58]. To solve this problem, some post-processing methods [34, 42, 74] have been proposed, which generally use additional denoising tools to reduce

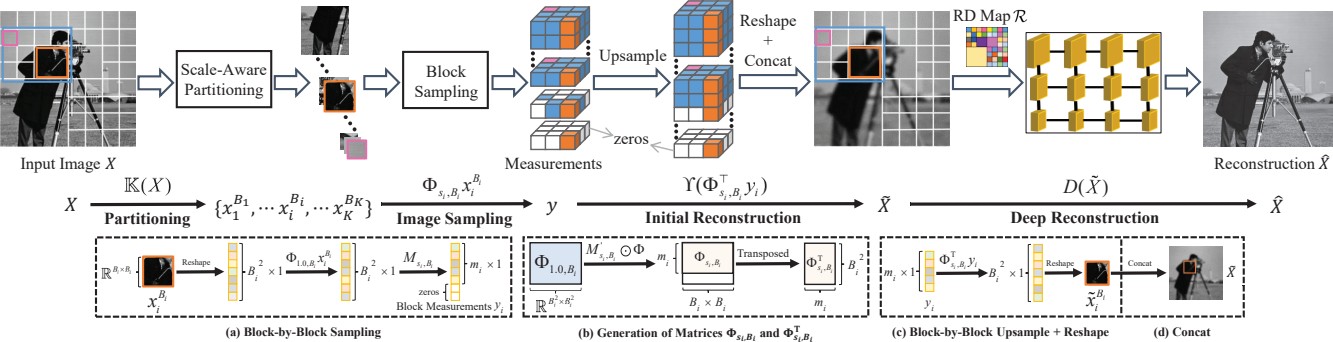

**Figure 1:** Diagram of our proposed S²-CSNet, which consists of a scale-aware partitioning sub-module, an block sampling sub-module, an initial reconstruction sub-module, and a deep reconstruction sub-module. (a)-(d) presents some implementation details, including (a) the process of block-by-block sampling, (b) the generation of the submatrices, and (c) (d) the operations of upsample, reshape, and concat in the initial reconstruction sub-module.

the blocking artifacts. In addition, some literatures [7, 57, 58, 63] attempt to explore global deep prior, which concatenates all measurements into a complete image and feeds it into a well-designed network for optimization. Recently, some deep unfolded networks (DUNs) have been applied to image CS to provide better theoretical basis and inferential interpretability [5, 60, 61, 78–80, 84]. Specifically, DUN usually unfolds certain optimization solvers into deep network forms [4, 59, 69, 75], such as multi-scale block CS (MS-BCS) algorithm [18], proximal gradient descent (PGD) algorithm [52] and approximate message passing (AMP) algorithm [11]. In general, none of the aforementioned deep network-based image CS methods consider improving performance at the scale-aware sampling level, which leads to limited insights into the image content.

## 3 Proposed Method

### 3.1 Overview of S²-CSNet

Fig. 1 shows the algorithmic workflow and the main structure of S²-CSNet. Considering the diversity of image contents, a scale-aware adaptive partitioning (SAP) sub-module divides the image into sub-blocks of multiple scales. Specifically, in SAP, to handle the input image block with arbitrary possible sampling rates and different scales, a universal sampling matrix is first designed, which is a structural sampling matrix with a predefined order. Based on this sampling matrix, a distortion-guided scale-aware scheme is presented, which can evaluate the reconstruction distortion (RD) of the image under different scales of sampling and select the optimal partitioning manner. In the sampling sub-module, image blocks of different scales are sampled with arbitrary assigned sampling rates to obtain the compressed measurements (see Eq. (2)). In the reconstruction sub-module, an initial reconstruction sub-network and a multi-scale hierarchical deep reconstruction sub-network with a dual spatial attention mechanism (see Fig. 3) are proposed to reconstruct the original image from the sampled measurements.

### 3.2 Scale-Aware Scalable Sampling

This section describes the implementation details of the scale-aware scalable sampling. Specifically, given the input image $X$ and the overall sampling rate $R$, it will be divided into blocks of multiple scales, i.e., $\{(x_1^{B_1}, s_1), \cdots, (x_i^{B_i}, s_i), \cdots, (x_K^{B_K}, s_K)\}$, where $s_i$ is the sampling rate assigned to each block based on the reconstruction distortion prior and $K$ is the total number of blocks.

*1) Design of the universal sampling matrix.* In scale-aware sampling, a specific-size sampling matrix $\Phi_{s_i,B_i} \in \mathbb{R}^{m_i \times lB_i^2}$ is designed to perform the sampling operation for image blocks $x_i^{B_i}$ with scale $B_i$ and sampling rate $s_i$. In order to handle arbitrary sampling rates, the non-adaptive BCS methods need to train $lB^2$ sampling matrices [5, 29]. If there are $q$ variable scales, then $qlB^2$ sampling matrices are required to realize S²-CSNet. To reduce the number of sampling matrices, like [5, 29, 80], we propose to design a universal sampling matrix $\Phi_{1.0,\mathbb{B}} = \{\Phi_{1.0,B_1}, \Phi_{1.0,B_2}, ..., \Phi_{1.0,B_{max}}\}$ to handle multiple scales $\mathbb{B} = \{B_1, B_2, \cdots, B_{max}\}$ and arbitrary sampling rates. For a block $x_i^{B_i}$, the sampling process can be formulated as follows:

$$\begin{cases} y_i = \Phi_{s_i,B_i} x_i^{B_i} = (\Phi_{1.0,B_i} x_i^{B_i}) \odot M_{s_i,B_i} \\ s.t. \quad s_i = \mathcal{G}(y, \Phi_{1.0,B_i}, R) \end{cases} \quad (2)$$

where $M_{s_i,B_i} \in \mathbb{R}^{K \times lB_i^2}$ is a measurement mask used to extract the measurements at sampling rate $s_i$ (details are shown in Fig. 1), $\odot$ is the element-wise product, and $\mathcal{G}$ is a sampling rate allocation function. It should be noted that $\mathcal{G}$ is not required for the original image to guide the sampling rate allocation, which is related to Eq. (4) in Section 3.2.2, i.e., $\mathcal{G}$ allocates the sampling rate based on the distortion calculated by Eq. (4). Furthermore, to fully train the sampling matrix to handle arbitrary sampling rates, we define a decreasing trend of measurement base (i.e., each row of the matrix) importance from the first row to the last row [5]. In terms of handling sampling at multiple scales, the different sub-sampling matrices in $\Phi_{1.0,\mathbb{B}}$ are restricted to satisfy different scales of sampling by solving the following optimization problem:

$$\underset{\Phi_{1.0,\mathbb{B}}}{\arg\min} \sum_{j=1}^{N_s} \sum_{B_i=B_1}^{B_{max}} ||D[I(\Phi_{1.0,B_i} X_j)] - X_j||_2^2, B_i \in \mathbb{B} \quad (3)$$

where $I$ and $D$ are the initial and deep reconstruction, respectively, $N_s$ is the number of samples and $X_j$ is a validation sample.

*2) Scale-aware adaptive sampling.* The difficulty of CS reconstruction varies depending on the image content [29], while existing BCS mechanically divides the image by a fixed scale, which leads to the mixing of information with different content characteristics and affects the performance of CS sampling. To solve this problem, we propose a scale-aware sampling strategy, which can efficiently distinguish the image content according to the RD prior of the image. In this subsection, we will describe both the calculation of the RD prior and RD-based scale-aware sampling.

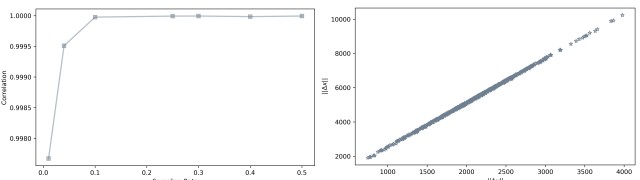

**Figure 2:** The correlation between $||\Delta x||$ and $||\Delta y||$ at different sampling rates (left) and the statistical diagram of $||\Delta x||$ and $||\Delta y||$ at $R = 0.25$ (right).

**Calculation of the RD prior**: During the actual sampling process of CS, we cannot fully acquire the input image in some cases [5, 29]. In order to obtain the content properties of the original image, some ideal calculation methods, such as [80, 88], which employ multiple detectors to compute the salience directly on the original image. Different from these methods, inspired by [29, 53], we propose a distortion estimation method in the measurement domain that only uses the sampling matrix to perfom linear operations on the measurements, which can be defined as:

$$
\begin{cases}
||\Delta y_i^{B_i}|| = ||\Phi_{R_{bs}, B_i} \Delta x_i^{B_i}|| = ||\Phi_{R_{bs}, B_i} x_i^{B_i} - \Phi_{R_{bs}, B_i} \tilde{x}_i^{B_i}|| \\
\quad = ||y_i - (M'_{R_{bs}, B_i} \odot \Phi_{1.0, B_i}) \tilde{x}_i^{B_i}||
\end{cases}
\tag{4}
$$

where $\tilde{x}_i^{B_i}$ is the initial reconstruction recovered from the measurements $y_i$ of the image block $x_i^{B_i}$ (see Section 3.3 for details), $\Phi_{R_{bs}, B_i}$ is a submatrix truncated from $\Phi_{1.0, B_i}$ with a matrix mask $M'_{R_{bs}, B_i}$, $R_{bs}$ is the base sampling rate, which is usually a portion of $R$. $\Delta y_i^{B_i}$ is the estimation of the real reconstruction distortion of the image block $x_i^{B_i}$, and $|| \cdot ||$ is the absolute operator. It is worth noting that the sampling of $S^2$-CSNet consists of two parts, the base sampling for Eq. (4) and the scale-aware sampling for Eq. (2). In Eq. (2), if a mask $M'$ is used to generate the sub-matrix $\Phi_{s_i, B_i}$ for sampling, $S^2$-CSNet will require two times of sampling operations. Instead, Eq. (2) only requires one sampling operation, and the measurements are used for both the base sampling and scale-aware sampling. To verify the accuracy of predicting $||\Delta x||$ from $||\Delta y||$, Fig. 2 shows the correlation [26, 28] and the statistical diagram of these two variables. It can be observed that they are approximately linearly correlated, i.e., the bigger the $||\Delta x||$, the bigger the corresponding $||\Delta y||$. Therefore, it is plausible to deduce $||\Delta x||$ from $||\Delta y||$.

**Scale-aware sampling**: As shown in Fig. 1, scale-aware scalable sampling consists of two steps: scale-aware partitioning and adaptive block sampling. Given an image $X \in \mathbb{R}^{H \times W}$, and a group of sampling scales $\mathbb{B} = \{B_1, B_2, \cdots, B_{\max}\}$ with $B_1 < B_2 < \cdots < B_{\max}$, we first compute the RD for each sampling scale $B_i$ based on Eq. (4):

$$
\mathcal{R}_{B_i} = \Psi(||y - (M'_{R_{bs}, B_i} \odot \Phi_{1.0, B_i}) \tilde{X}||)
\tag{5}
$$

where $\Psi$ is a repeat operation that expands the computed distortion value for each block into a two-dimensional matrix $\mathcal{R}_{B_i} \in \mathbb{R}^{(H \times W)}$ by filling in the distortion values at the corresponding locations of $X$. Then, the image $X$ is uniformly divided into multiple blocks $\{x_1^{B_{\max}}, x_2^{B_{\max}}, \cdots, x_K^{B_{\max}}\}$ based on the largest scale $B_{\max}$. In the aggregation process, a recursive comparison function $\Gamma$ will compare the distortion values of different scales in each sub-block $x_i^{B_{\max}}$ and return the scale $B_s$ with the minimum distortion:

$$
B_s = \Gamma(x_i^{B_{\max}}, \mathcal{R}_{B_1}, \mathcal{R}_{B_2}, \cdots, \mathcal{R}_{B_{\max}})
\tag{6}
$$

Specifically, $\Gamma$ compares all division schemes under sub-block $x_i^{B_{\max}}$, i.e., $x_i^{B_{\max}}$ is partitioned into multiple sub-scales from $B_1$ to $B_{\max -1}$. Next, $\Gamma$ chooses the scale $B_s$ in $\mathbb{B}$ with the minimum distortion, i.e.,

$||\Delta y_i^{B_s}|| < ||\Delta y_i^{B_j}||$, where $B_j$ is an arbitrary scale and $B_j \neq B_s$. Taking the scale $\mathbb{B} = \{B_1 = 16, B_{\max} = 32\}$ as an example, the image is uniformly divided into $32 \times 32$ sub-blocks according to the maximum scale 32. For each sub-block $x_i^{B_{\max}}$, we compare whether to further divide it into 4 sub-blocks in a $16 \times 16$ manner or to keep the size of $32 \times 32$ to minimize the RD. The RD has been calculated using Eq. (4), and is recorded in the RD maps $\mathcal{R}_{B_1}$ and $\mathcal{R}_{B_{\max}}$. Therefore, it is sufficient to extract the distortion values at the position of $x_i^{B_{\max}}$ from the $\mathcal{R}_{B_i}$ of the corresponding scale. Then, $\Gamma$ will compare the distortions under all division modes (i.e., $B_1$ and $B_{\max}$) of $x_i^{B_{\max}}$ and choose a scale with the smallest distortion as the optimal division scale. Finally, we mark the corresponding distortion value to the position of $x_i^{B_{\max}}$ in RD map $\mathcal{R}$, and perform the CS sampling $y_i = \Phi_{s_i, B_s} x_i^{B_{\max}}$.

## 3.3 Reconstruction with Dual Spatial Attention

*1) Initial reconstruction.* For the sampled measurements $y_i \in \mathbb{R}^{m_i \times 1}$ of an image block $x_i^{B_i}$, there are usually two stages of reconstruction, including an initial reconstruction and a deep reconstruction [5, 58, 88]. Specifically, $y_i$ will first be upsampled (i.e., $\Phi_{s_i, B_i}^\top y_i$) to produce a $B_i^2 \times 1$ vector. Next, a reshape operation ($\Upsilon$) is used to transform all vectors into $B_i \times B_i$ tensor blocks (i.e., $\tilde{x}_i^{B_i}$). Fig. 1 illustrates the flow of the detailed initial reconstruction for $\tilde{x}_i^{B_i}$, which can be summarized as $\tilde{x}_i^{B_i} = \Upsilon(\Phi_{s_i, B_i}^\top y_i)$. At last, all tensor blocks are concatenated to output the complete reconstructed image $\tilde{X}$.

*2) Dual spatial attention-based deep reconstruction.* Since $\tilde{X}$ is a coarse reconstruction, a dual spatial attention-based hierarchical network is proposed to finely optimize $\tilde{X}$. The reconstruction process for $\tilde{X}$ consists of two parts of inputs, one is the basis measurements $y_{bs}$ used to compute the RD (see Section 3.2.2), and the other is the remaining measurements $y_{es}$ from the adaptive sampling. The overall measurements $y$ are composed of $y_{bs}$ and $y_{es}$, and with the exploitation of RD information [5], the reconstruction process $\hbar(y)$ can be represented as:

$$
\hbar(y) = D[I(y_{es} \cup y_{bs}) \,|\, \varepsilon(\mathcal{R})]
\tag{7}
$$

where $\varepsilon$ is a feature extractor consisting of several convolutions and residual blocks [5], $|$ is the operation that concatenates $I(\cdot)$ and $\varepsilon(\cdot)$, and $\cup$ denotes the union of sets. Fig. 3 illustrates the overall framework of the deep reconstruction network, which involves a series of horizontal and vertical branches to construct a hierarchical grid architecture [7, 17, 29]. Specifically, the horizontal branch consists of multiple deep reconstruction blocks (DRBs), which are responsible for feature extraction and local and global affinity mining in a certain scale space. The vertical branch is composed of upsampling and downsampling submodules, which are used to integrate the intermediate feature maps of different horizontal branches. For a DRB positioned in the $i$-th row and $j$-th column, its output feature map $\breve{x}_{i,j}$ can be summarized as the following two equations:

$$
\breve{x}_{i,j} = \Theta_{i,j}^{DRB}[\breve{x}_{i,j-1} \oplus \Theta_{i-1,j-1}^{DS}(\breve{x}_{i-1,j-1}, \theta_{i-1,j-1}^{DS}), \theta_{i,j}^{DRB}]
\tag{8}
$$

$$
\breve{x}_{i,j} = \Theta_{i,j}^{DRB}[\breve{x}_{i,j-1} \oplus \Theta_{i+1,j-1}^{US}(\breve{x}_{i+1,j-1}, \theta_{i+1,j-1}^{US}), \theta_{i,j}^{DRB}]
\tag{9}
$$

where $\Theta^{DRB}$, $\Theta^{DS}$ and $\Theta^{US}$ denote the mapping operations of DRB, downsampling submodule and upsampling submodule, respectively, and $\theta$ is the trainable parameter. It should be noted that Eq. (8)

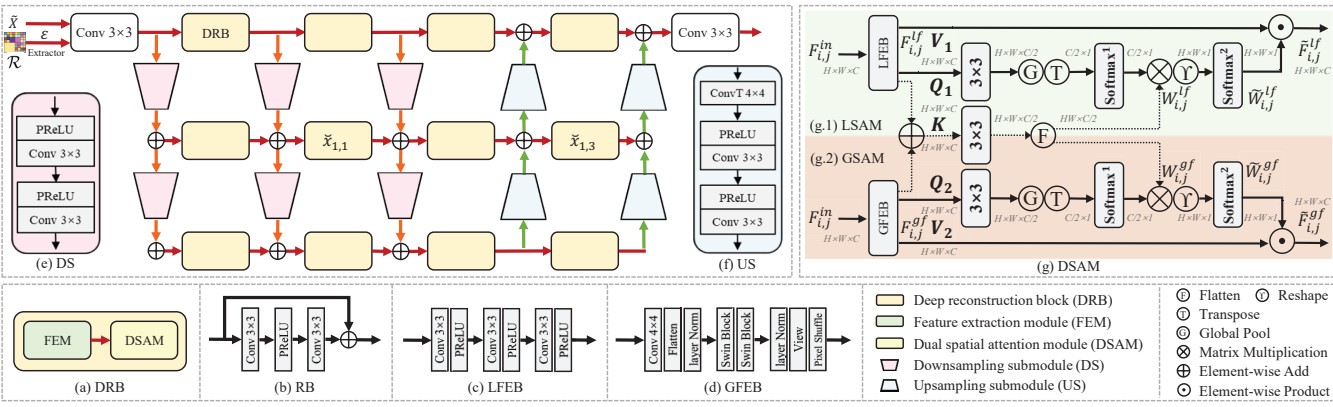

**Figure 3:** Details of the deep reconstruction network structure, which is a grid architecture consisting of multiple horizontal and vertical branches. Specifically, the horizontal branch contains several cascaded DRBs, and each DRB consists of FEM (including a few residual blocks (RB)) and DSAM (including LSAM and GSAM), which are responsible for feature extraction and local global attention modeling, respectively. The DSAM uses the spatial attention mechanism to fuse the local features extracted by LFEB and the global features extracted by GFEB to enhance the reconstruction ability. For vertical branches, a number of downsampling and upsampling sub-branches are employed to boost the interaction of the horizontal branches. The configuration of each sub-module is shown as (a)-(g).

mainly integrates downsampled features and Eq. (9) mainly integrates upsampled features, e.g. they can calculate the outputs $\breve{x}_{1,1}$ and $\breve{x}_{1,3}$ of the DRBs, respectively.

In each DRB, we design two main parts, one is the feature extraction module (FEM), which is composed of several residual blocks. The other is the dual spatial attention module (DSAM), which is employed to model the local features $F_{i,j}^{lf}$ extracted by the local feature extraction block (LFEB) and the global prior features $F_{i,j}^{gf}$ extracted by the global feature extraction block (GFEB). DSAM contains local spatial attention module (LSAM) and global spatial attention module (GSAM), and they use spatial attention (SA) mechanism to compute the weights of $F_{i,j}^{lf}$ and $F_{i,j}^{gf}$ respectively to deeply fuse local and global features. Specifically, for the inputs Q and K of SA, a 3×3 convolution is first used to extract the feature to generate Q'$\in\mathbb{R}^{H\times W\times(C/2)}$ and K'$\in\mathbb{R}^{H\times W\times(C/2)}$. To get the input tokens $\{\tilde{Q},\tilde{K},\tilde{V}\}$, a global pooling is applied to reduce the dimensionality of Q' to get $\tilde{Q}$. For K', it is flattened into the dimension of HW×(C/2) to obtain $\tilde{K}$. In addition, the input V$\in\mathbb{R}^{H\times W\times C}$ is left unchanged to generate $\tilde{V}$.

Next, we use the softmax function to reweight $\tilde{Q}^\top$ and conduct matrix multiplication with $\tilde{K}$ to generate the transposed attention map $W_{i,j}\in\mathbb{R}^{HW\times1}$, i.e., $W_{i,j}=\tilde{K}\otimes\text{Softmax}^1(\tilde{Q}^\top)$. Here, $W_{i,j}$ actually represents the weight of the feature $\tilde{V}$, which is calculated from the query feature map Q and the key feature map K. To match the dimension of $\tilde{V}$, we reshape $W_{i,j}$ to size H×W×1 and use Softmax2D (i.e., Softmax$^2$ in Fig. 3) to generate the normalized weights $\tilde{W}_{i,j}$. The weighted aggregation of $\tilde{W}_{i,j}$ and $\tilde{V}$ can be computed as:

$$\mathcal{A}_{SA}(Q,K,V) = \text{Softmax}^2(\Upsilon(W_{i,j})) \odot \tilde{V} \qquad (10)$$

In LSAM (i.e., (g.1) in Fig. 3) and GSAM (i.e., (g.2) in Fig. 3), the inputs Q, K and V of SA can be respectively defined as:

$$V_{i,j}^{LF}, Q_{i,j}^{LF}, K_{i,j}^{LF} = F_{i,j}^{lf}, F_{i,j}^{lf}, (F_{i,j}^{lf} + F_{i,j}^{gf}) \qquad (11)$$

$$V_{i,j}^{GF}, Q_{i,j}^{GF}, K_{i,j}^{GF} = F_{i,j}^{gf}, F_{i,j}^{gf}, (F_{i,j}^{lf} + F_{i,j}^{gf}) \qquad (12)$$

where the feature map K is a rough fusion of $F_{i,j}^{lf}$ and $F_{i,j}^{gf}$. At last, the output feature $F_{i,j}^{out}$ of DSAM can be represented as:

$$\begin{cases} F_{i,j}^{out} = \tilde{W}_{i,j}^{lf} \odot F_{i,j}^{lf} + \tilde{W}_{i,j}^{gf} \odot F_{i,j}^{gf} \\ = \mathcal{A}_{SA}(Q_{i,j}^{LF}, K_{i,j}^{LF}, V_{i,j}^{LF}) + \mathcal{A}_{SA}(Q_{i,j}^{GF}, K_{i,j}^{GF}, V_{i,j}^{GF}) \end{cases} \qquad (13)$$

where $\tilde{F}_{i,j}^{lf}=\tilde{W}_{i,j}^{lf}\odot F_{i,j}^{lf}$ and $\tilde{F}_{i,j}^{gf}=\tilde{W}_{i,j}^{gf}\odot F_{i,j}^{gf}$ are the outputs of LSAM and GSAM, respectively.

## 3.4 Loss Function

In the training of S$^2$-CSNet, the learnable parameters include the sampling matrix $\Phi_{1.0,\mathbb{B}}$ and the reconstruction network. Given the training set $\{X_j\}_{j=1}^{N_p}$, we employ the following $\ell_2$-loss like [58] to train S$^2$-CSNet, which can be defined as:

$$\mathcal{L}(\Phi_{1.0,\mathbb{B}}, \theta_d) = \frac{1}{N_p} \sum_{j=1}^{N_p} \sum_{B_i=B_1}^{B_{\max}} ||\mathcal{F}(\Phi_{s_i,B_i}X_j, n_j, \theta_d) - X_j||_2^2 \quad (14)$$

where $\theta_d$ is the parameter of the reconstruction network, $N_p$ is the number of training images, $n_j$ is a non-negative integer, and the output of $\mathcal{F}(\cdot)$ is the reconstructed image $\hat{X}_j$. In the input of $\mathcal{F}$, $n_j$ is a randomly generated integer from $\{1,2,\cdots,Q'\}$, which is utilized to define the number of measurements, i.e., $s_j=n_j/Q'$, where $Q'=lB_{\max}^2$. In each training epoch, a random $n_j$ is selected to train the ability of the sampling matrix to cope with arbitrary sampling rates [5]. Furthermore, during the optimization process, Eq. (14) ensures the training of sampling matrices for different scales in $\Phi_{1.0,\mathbb{B}}$ by partitioning the image into different scales.

## 4 Experiments and Analysis

### 4.1 Implementation and Training Details

In block-based CS methods [5, 58, 61, 80], the block size is usually set to 16 or 32. Following these works, in the scale-aware sampling of the proposed S$^2$-CSNet, we set the number of sampled scales to 2, i.e., $\mathbb{B} = \{16, 32\}$. We implement S$^2$-CSNet with PyTorch on two NVIDIA RTX 3060 GPUs, employ Adam [32] optimizer with a batch size of 20. S$^2$-CSNet is trained with a learning rate of 1e-4 for 300 epochs, and as the number of epoch increases, the learning rate

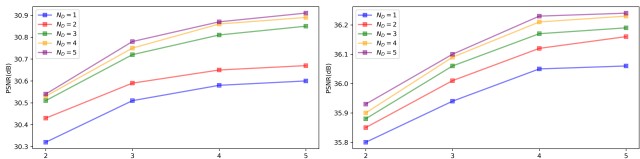

**Figure 4:** The relationship between hyperparameters ($N_B$, $N_D$) and the reconstruction quality at different sampling rates. *Left:* $R = 0.1$; *Right:* $R = 0.25$.

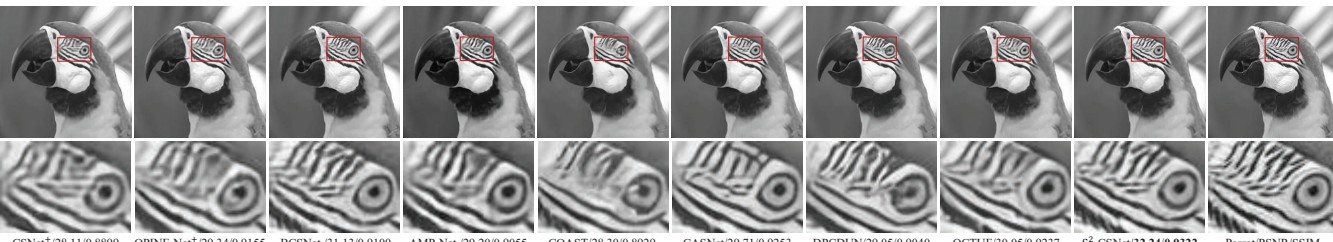

CSNet⁺/28.11/0.8899  OPINE-Net⁺/29.34/0.9155  BCSNet /31.13/0.9100  AMP-Net /29.20/0.9055  COAST/28.30/0.8929  CASNet/29.71/0.9253  DPCDUN/29.05/0.9040  OCTUF/30.05/0.9237  S²-CSNet/**32.24/0.9322**  Parrot/PSNR/SSIM

**Figure 5:** Visual comparisons of our S²-CSNet and other representative image CS networks on recovering an image from Set11 [34] in the case of CS ratio $R = 0.10$.

is multiplied by a factor of 0.5 every 50 epochs. The training set is the same as [5], which contains 25600 randomly cropped 128×128 image patches from T91 [9] and Train400 [81]. For testing, we utilize two widely used datasets: Set11 [34] and CBSD68 [46]. The sampling rate allocation is similar to [88], supposing that $v$ denotes the amount of distortion information embodied in image $X$. One has $v = \frac{1}{n} \sum_{j \in \mathcal{R}} l_j$, where $n$ is the total number of pixels on image $X$ and $l_j$ is the distortion value of location $j$ on RD map $\mathcal{R}$. For block $x_i$ of image $X$, its sampling rate can be calculated as $s_i = \frac{v_i}{v} \times R$, where $v_i$ represents the distortion information of $x_i$, and $R$ is the given sampling rate. In addition, two Swin Transformer Blocks [41] are used in the GFEB of the deep reconstruction network, and the number of multi-head self-attention block is set to 4, the shift distance is set to 4 and the window size is set to 8.

## 4.2 Exploration of Model Hyperparameters

*1) Exploring the architecture of deep reconstruction network in Fig. 3:* We first explore the setting of the number of horizontal branches ($N_B$) and the number of DRBs ($N_D$) on each horizontal branch. Fig. 4 reveals the relationship between these two hyperparameters and the image reconstruction quality, from which it can be seen that as these two hyperparameters increase, the reconstruction quality becomes less and less sensitive to them. Therefore, we set $N_B = 4$ and $N_D = 4$. Correspondingly, the number of sets of vertical branches is set to 5, including 3 sets of downsampling branches and 2 sets of upsampling branches as displayed in Fig. 3. To simplify the model and reduce complexity, we apply DSAM only in the first horizontal branch. On this branch, each DRB contains a set of FEM and DSAM, with the number of RBs in FEM set to 6. The DRBs of the remaining horizontal branches consist of RBs, and the number of RBs contained in the DRBs of the different branches is set to 2, 3 and 4, respectively. In addition, we set the number of input and output channels of DRB to 64.

*2) Setting of the base sampling rate to acquire the RD:* Scale-aware partitioning relies on the RD prior, in Eq. (4), we set a small base sampling rate $R_{bs}$ to compute the reconstruction distortion of the image. $R_{bs}$ is part of the overall sampling rate $R$, i.e., $R_{bs} = \gamma \times R$. To test the image reconstruction quality under different proportion

coefficients $\gamma$, Fig. 6 shows the average PSNR curve on Set11 and CBSD68 at seven sampling rates. It can be observed that when $\gamma$ approaches 0 or 1, S²-CSNet degrades into a uniform sampling version due to the lack of prior computation or sufficient space for adaptive sampling rate allocation. Therefore, we set $\gamma = 0.15$, where it reaches the peak of reconstruction quality.

*3) Analysis of the learned universal sampling matrix $\Phi_{1.0,\mathbb{B}}$:* Like [5, 29], we analyze the properties of the learned sampling matrix from three aspects, including: orthogonality, coefficient distribution, and frequency view. For orthogonality: It can be seen from Fig. 8 that the property $\Phi\Phi^\top = \mu I$ is approximately satisfied for both $B = 16$ and $B = 32$ sampling matrices without additional constraints, where I is the identity matrix. For coefficient distribution: We show the histograms of the learned matrices and Gaussian random matrices in Fig. 8. It can be observed that the learned matrices exhibit a wider and sparser distribution. For the frequency view: we reshape one row of the sampling matrix to $16 \times 16$ and $32 \times 32$, respectively, and present their spatial and frequency domain views. Fig. 9 shows the views of the first ten rows of the learned sampling matrices, indicating that they exhibit structured and anisotropic spatial patterns different from traditional manually defined filters. In addition, the rows at the front of the sampling matrix have narrower frequencies, which implies that they pay more attention to low-frequency information at low sampling rates.

## 4.3 Comparisons with Other Methods

*1) Overall Comparisons:* In Table 3, we compare S²-CSNet with thirteen representative state-of-the-art CS algorithms. Here, we categorize these methods into two groups: CS with adaptive CS ratio allocation (ACRA) and CS with fixed CS ratio allocation. For CS based on ACRA, such as BCSNet [88], AMSNet [80], and CASNet [5], they adaptively allocate sampling rates based on the content of the image. However, methods like [88] and [80] require the original image to compute prior features to guide the allocation of sampling rates. Furthermore, BCSNet [88] only implements a limited number of sampling rate allocations, lacking scalability.

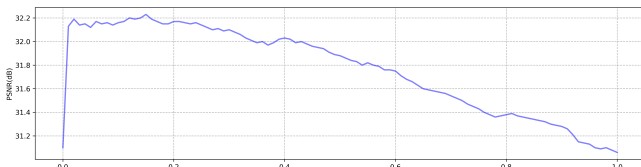

**Figure 6:** Average PSNR curve on Set11 and CBSD68 with CS sampling rates $R \in \{0.01, 0.04, 0.10, 0.25, 0.30, 0.40, 0.50\}$, which performs best at $\gamma = 0.15$.

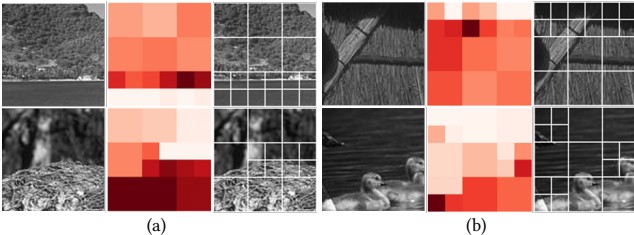

(a)        (b)

**Figure 7:** Visualization of two sets of scale-aware sampling (right) and scale-variable adaptive sampling rate allocation (shown as the heatmap in the middle) with the corresponding original image (left).

**Table 1: Average PSNR(dB) and SSIM comparisons of recent deep network-based CS algorithms on the Set11 dataset. The best performances are highlighted in bold and the second best performances are indicated by underlining.**

| Dataset | Algorithms | Rate=0.01 | | Rate=0.04 | | Rate=0.10 | | Rate=0.25 | | Rate=0.30 | | Rate=0.40 | | Rate=0.50 | | Avg. | |
|---|---|---|---|---|---|---|---|---|---|---|---|---|---|---|---|---|---|
| | | PSNR | SSIM | PSNR | SSIM | PSNR | SSIM | PSNR | SSIM | PSNR | SSIM | PSNR | SSIM | PSNR | SSIM | PSNR | SSIM |
| Set11 [34] | ISTA-Net$^+$(CVPR2018)[78] | 17.48 | 0.4479 | 21.32 | 0.6037 | 26.64 | 0.8087 | 32.59 | 0.9254 | 33.68 | 0.9352 | 35.97 | 0.9544 | 38.11 | 0.9707 | 29.40 | 0.8066 |
| | SCSNet(CVPR2019)[57] | 21.04 | 0.5562 | 24.29 | 0.7589 | 28.52 | 0.8616 | 33.43 | 0.9373 | 34.64 | 0.9511 | 36.92 | 0.9666 | 39.01 | 0.9769 | 31.12 | 0.8584 |
| | DPA-Net(TIP2020)[63] | 18.05 | 0.5011 | 23.50 | 0.7205 | 26.99 | 0.8354 | 31.74 | 0.9238 | 33.35 | 0.9425 | 35.21 | 0.9580 | 36.80 | 0.9685 | 29.38 | 0.8357 |
| | CSNet$^+$(TIP2020)[58] | 20.67 | 0.5411 | 24.83 | 0.7480 | 28.34 | 0.8580 | 33.34 | 0.9387 | 34.27 | 0.9492 | 36.44 | 0.9690 | 38.47 | 0.9796 | 30.91 | 0.8548 |
| | OPINE-Net$^+$(JSTSP2020)[79] | 20.15 | 0.5340 | 25.69 | 0.7920 | 29.81 | 0.8904 | 34.86 | 0.9509 | 35.79 | 0.9541 | 37.96 | 0.9633 | 40.19 | 0.9800 | 32.06 | 0.8664 |
| | BCSNet(TMM2020)[88] | 20.86 | 0.5510 | 24.90 | 0.7531 | 29.42 | 0.8673 | 34.20 | 0.9408 | 35.63 | 0.9495 | 36.68 | 0.9667 | 39.58 | 0.9734 | 31.61 | 0.8574 |
| | AMP-Net(TIP2021)[84] | 20.55 | 0.5638 | 25.14 | 0.7701 | 29.42 | 0.8782 | 34.60 | 0.9469 | 35.91 | 0.9576 | 38.25 | 0.9714 | 40.26 | 0.9786 | 32.02 | 0.8667 |
| | COAST(TIP2021)[76] | - - | - - | - - | - - | 30.03 | 0.8946 | - - | - - | 36.35 | 0.9618 | - - | - - | 40.32 | 0.9804 | - - | - - |
| | AMSNet(TMM2022)[80] | 21.51 | 0.5772 | 26.32 | 0.7951 | 30.45 | 0.8823 | 35.76 | 0.9426 | 37.15 | 0.9583 | 39.26 | 0.9602 | 40.95 | 0.9734 | 33.06 | 0.8699 |
| | CASNet(TIP2022)[5] | _21.97_ | _0.6140_ | _26.41_ | _0.8153_ | 30.36 | 0.9014 | 35.67 | 0.9591 | 36.92 | 0.9662 | 39.04 | 0.9760 | 40.93 | 0.9826 | 33.04 | _0.8878_ |
| | DPC-DUN(TIP2023)[60] | 18.12 | 0.4785 | 24.39 | 0.7501 | 29.40 | 0.8798 | 34.69 | 0.9482 | 35.88 | 0.9570 | 37.98 | 0.9694 | 39.84 | 0.9778 | 31.47 | 0.8515 |
| | CAT-Net(TMM2023)[33] | 21.29 | 0.5782 | 26.38 | 0.8060 | 30.69 | 0.9022 | 35.85 | 0.9588 | 37.12 | 0.9668 | 39.32 | 0.9766 | 41.28 | 0.9834 | _33.13_ | 0.8817 |
| | OCTUF(CVPR2023)[61] | - - | - - | - - | - - | _30.70_ | _0.9030_ | _36.10_ | _0.9604_ | _37.21_ | _0.9673_ | _39.41_ | _0.9773_ | **41.34** | _0.9838_ | - - | - - |
| | S²-CSNet (Ours) | **22.57** | **0.6183** | **26.95** | **0.8186** | **30.86** | **0.9045** | **36.21** | **0.9614** | **37.33** | **0.9681** | **39.47** | **0.9785** | _41.31_ | **0.9846** | **33.53** | **0.8906** |

CASNet [5] employs a universal sampling matrix to organically achieve both model scalability and ACRA. Overall, there is currently no algorithm that considers scale-aware adaptive sampling.

*2) Comparisons of Reconstruction Quality:* Visual quality is an important measure of the algorithm [22–25, 38, 47–49]. In Table 1 and Table 2, we present the comparison of different image CS methods on the Set11 and CBSD68 datasets in terms of PSNR and SSIM metrics. Compared with representative non-ACRA methods (CSNet$^+$ [58], AMP-Net [84] and CAT-Net [33]), a) on the dataset Set11, the proposed S²-CSNet achieves on average 2.62dB, 1.51dB, 0.40dB and 0.0358, 0.0239, 0.0089 gains in PSNR and SSIM compared against these three methods at the given sampling rates. b) On the dataset CBSD68, our proposed algorithm achieves on average 0.76dB, 0.69dB, 0.51dB and 0.0265, 0.0303, 0.0123 gains in PSNR and SSIM under different sampling rates. Compared with representative ACRA methods (AMSNet [80] and CASNet [5]), a) On the dataset Set11, the proposed S²-CSNet achieves on average 0.47dB, 0.49dB and 0.0207, 0.0028 gains in PSNR and SSIM compared with these ACRA methods under the given sampling rates. b) On the dataset CBSD68, our proposed method achieves on average 0.16dB, 0.10dB and 0.0099, 0.0011 gains in PSNR and SSIM in terms of different sampling rates. The visual comparisons in Fig. 5 shows that our S²-CSNet is able to recover high-quality results with more details.

*3) Comparisons of Complexity:* The computation cost and model parameters are important in many practical applications [36, 64, 65]. To verify the efficiency of the proposed S²-CSNet, in Table 3, we compare the number of parameters of different CS methods and the speed of sampling and reconstructing a 256×256 image at sampling rates $R \in \{0.01, 0.04, 0.10, 0.25, 0.30, 0.40, 0.50\}$ on a 1080Ti GPU. Due to the additional prior computation and adaptive sampling rate allocation required by ACRA-based CS (e.g., AMSNet [80], CASNet [5]), their speed is generally slower than non-ACRA-based CS (e.g., CSNet$^+$ [58], AMP-Net [84]). Compared with recent ACRA-based methods (AMSNet [80] and CASNet [5]), the proposed S²-CSNet basically maintains a similar complexity with them. Specifically, S²-CSNet has fewer parameters than CASNet [5] but higher than AMSNet [80]. In terms of running speed, S²-CSNet is faster than AMSNet [80] and CASNet [5].

## 4.4 Ablation Studies

*1) Effect of Scale-Aware Adaptive Partitioning (SAP):* Fig. 7 displays some examples of scale-aware sampling, and it can be seen that the proposed S²-CSNet can effectively distinguish image contents and achieve finer sampling rate allocation. Specifically, we observe that S²-CSNet tends to use large-scale blocks for sampling in content-consistent regions such as the background, which will allow the convolution kernel to obtain a broader field of view in convolution-based sampling. In regions with complex content such as boundaries, small sampling scales can separate different content features more efficiently. Furthermore, in Fig. 9, we note that the frequency domain distribution of the sampling matrix with scale $16 \times 16$ is wider than that of the sampling matrix with scale $32 \times 32$, which suggests that the sampling matrix with scale $16 \times 16$ is more adept at processing high-frequency information. This is consistent

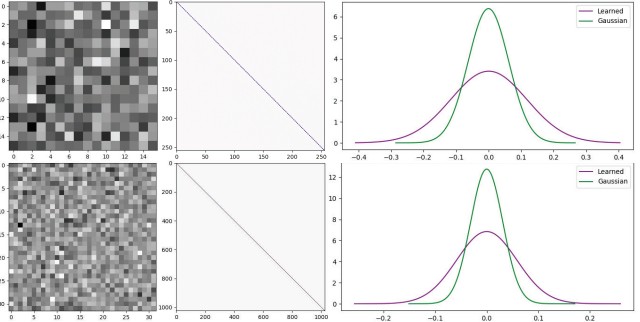

**Figure 8:** Comparison of the learned sampling matrices at two scales in terms of spatial view (left), coefficient distribution (right), and orthogonality (middle). The first row represents the sampling matrix with scale $B = 16$, while the second row represents the sampling matrix with scale $B = 32$.

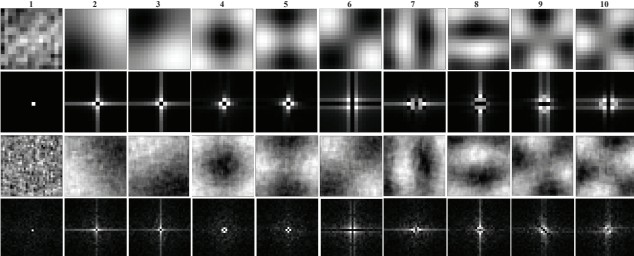

**Figure 9:** Visualization of first ten rows (1-10) selected from the learned sampling matrix, where the top two rows denote the matrix with scale $B = 16$ and the bottom two rows represent the matrix with scale $B = 32$. *1st and 3rd rows:* Spatial view; *2nd and 4th rows:* Frequency view.

**Table 2: Average PSNR(dB) and SSIM comparisons of recent deep network-based CS algorithms on the CBSD68 dataset. The best performances are highlighted in bold and the second best performances are indicated by underlining.**

| Dataset | Algorithms | Rate=0.01 | | Rate=0.04 | | Rate=0.10 | | Rate=0.25 | | Rate=0.30 | | Rate=0.40 | | Rate=0.50 | | Avg. | |
|---|---|---|---|---|---|---|---|---|---|---|---|---|---|---|---|---|---|
| | | PSNR | SSIM | PSNR | SSIM | PSNR | SSIM | PSNR | SSIM | PSNR | SSIM | PSNR | SSIM | PSNR | SSIM | PSNR | SSIM |
| CBSD68 [46] | ISTA-Net$^+$$_{(CVPR2018)}$ [78] | 19.14 | 0.4158 | 22.17 | 0.5486 | 25.32 | 0.7022 | 29.36 | 0.8525 | 30.25 | 0.8781 | 32.30 | 0.9195 | 34.04 | 0.9424 | 27.51 | 0.7513 |
| | SCSNet$_{(CVPR2019)}$ [57] | 22.03 | 0.5126 | 25.37 | 0.6623 | 28.02 | 0.8042 | 31.15 | 0.9058 | 32.64 | 0.9237 | 35.03 | 0.9214 | 36.27 | 0.9593 | 30.07 | 0.8128 |
| | DPA-Net$_{(TIP2020)}$ [63] | 20.25 | 0.4267 | 23.50 | 0.6205 | 25.47 | 0.7372 | 29.01 | 0.8595 | 29.73 | 0.8827 | 31.17 | 0.9156 | 32.55 | 0.9386 | 27.38 | 0.7830 |
| | CSNet$^+$$_{(TIP2020)}$ [58] | 22.21 | 0.5100 | 25.43 | 0.6706 | 27.91 | 0.7938 | 31.12 | 0.9060 | 32.20 | 0.9220 | 35.01 | 0.9258 | 36.76 | 0.9638 | 30.09 | 0.8131 |
| | OPINE-Net$^+$$_{(JSTSP2020)}$ [79] | 22.11 | 0.5140 | 25.20 | 0.6825 | 27.82 | 0.8045 | 31.51 | 0.9061 | 32.35 | 0.9215 | 34.95 | 0.9261 | 36.35 | 0.9660 | 30.04 | 0.8172 |
| | BCSNet$_{(TMM2020)}$ [88] | 21.95 | 0.5119 | 25.44 | 0.6597 | 27.98 | 0.8015 | 31.29 | 0.8846 | 32.70 | 0.9301 | 35.14 | 0.9397 | 36.85 | 0.9682 | 30.19 | 0.8137 |
| | AMP-Net$_{(TIP2021)}$ [84] | 22.18 | 0.5207 | 25.47 | 0.6534 | 27.79 | 0.7853 | 31.37 | 0.8749 | 32.68 | 0.9291 | 35.06 | 0.9395 | 36.59 | 0.9620 | 30.16 | 0.8093 |
| | COAST$_{(TIP2021)}$ [76] | - - | - - | - - | - - | 27.92 | 0.8061 | - - | - - | 32.66 | 0.9256 | - - | - - | 36.43 | 0.9663 | - - | - - |
| | AMSNet$_{(TMM2022)}$ [80] | 22.43 | 0.5473 | 25.68 | 0.6921 | 28.36 | 0.8054 | 32.23 | 0.9124 | 33.37 | 0.9302 | 35.38 | 0.9536 | 37.40 | 0.9668 | 30.69 | 0.8297 |
| | CASNet$_{(TIP2022)}$ [5] | 22.49 | 0.5520 | 25.73 | 0.7079 | 28.41 | 0.8231 | 32.31 | 0.9196 | 33.40 | 0.9359 | 35.43 | 0.9581 | 37.48 | 0.9728 | 30.75 | 0.8385 |
| | DPC-DUN$_{(TIP2023)}$ [60] | 20.08 | 0.4682 | 23.79 | 0.6220 | 26.72 | 0.7558 | 30.59 | 0.8797 | 31.63 | 0.9018 | 33.55 | 0.9340 | 35.44 | 0.9557 | 28.83 | 0.7882 |
| | CAT-Net$_{(TMM2023)}$ [33] | 22.28 | 0.5289 | 25.29 | 0.6888 | 27.95 | 0.8077 | 31.88 | 0.9115 | 32.98 | 0.9300 | 35.01 | 0.9542 | 37.02 | 0.9701 | 30.34 | 0.8273 |
| | OCTUF$_{(CVPR2023)}$ [61] | - - | - - | - - | - - | 28.28 | 0.8177 | 32.24 | 0.9185 | 33.32 | 0.9348 | 35.35 | 0.9578 | 37.41 | 0.9729 | - - | - - |
| | S$^2$-CSNet (Ours) | **22.84** | **0.5568** | **26.18** | **0.7118** | **28.65** | **0.8265** | **32.56** | **0.9224** | **33.54** | **0.9397** | **35.59** | **0.9595** | **37.59** | **0.9739** | **30.85** | **0.8396** |

with the example in Fig. 7, i.e., in high-frequency regions with complex content such as texture, S$^2$-CSNet will use small-scale sampling to achieve better reconstruction results. In contrast, in regions with simple and consistent content, S$^2$-CSNet prefers to use large-scale sampling to reconstruct low-frequency information. In Table 4, we present the reconstruction results of S$^2$-CSNet without the SAP module, including fixed-scale sampling with $B = 16$ (1st row) and $B = 32$ (2nd row). It can be seen that SAP can bring average PSNR gains of 0.33dB and 0.39dB for S$^2$-CSNet compared with the above two fixed-scale sampling.

*2) Effect of Adaptive CS Ratio Allocation (ACRA):* As one of the main ideas of S$^2$-CSNet, the allocation scheme based on reconstruction distortion can perform adaptive CS ratio allocation for blocks of different scales. Compared with the fixed CS ratio allocation scheme, ACRA can more effectively integrate the allocation of the sampling ratio into the content characteristics of the image and bring better reconstruction quality. The effectiveness of ACRA is also verified in Table 4, from which it can be seen that the average PSNR under fixed CS ratio sampling is 0.44dB lower than the sampling under the ACRA strategy.

*3) Effect of DSAM in Deep Reconstruction Network:* Compared with the native multi-scale reconstruction network [7, 29], the introduction of DSAM aims to enhance the mining of local and

**Table 3: Comparisons of different deep network-based CS methods in terms of model properties, overall parameters of seven CS ratios, and average running speed for reconstructing a $256 \times 256$ image.**

| Algorithm | ACRA[1] | SAP[2] | FGS[3] | WAGTI[4] | #Param. (M)/Time (ms) |
|---|---|---|---|---|---|
| ISTA-Net$^+$$_{(CVPR2018)}$ [78] | ✗ | ✗ | ✗ | ✔ | 2.38/35.79 |
| SCSNet$_{(CVPR2019)}$ [57] | ✗ | ✗ | ✔ | ✔ | 0.80/62.54 |
| DPA-Net$_{(TIP2020)}$ [63] | ✗ | ✗ | ✗ | ✔ | 65.17/70.31 |
| BCSNet$_{(TMM2020)}$ [88] | ✔ | ✗ | ✗ | ✗ | 1.64/117.25 |
| CSNet$^+$$_{(TIP2020)}$ [58] | ✗ | ✗ | ✗ | ✔ | 4.35/41.23 |
| OPINE-Net$^+$$_{(JSTSP2020)}$ [79] | ✗ | ✗ | ✗ | ✔ | 4.35/48.22 |
| AMP-Net$_{(TIP2021)}$ [84] | ✗ | ✗ | ✗ | ✔ | 6.08/58.34 |
| COAST$_{(TIP2021)}$ [76] | ✗ | ✗ | ✔ | ✔ | 1.12/76.25 |
| AMSNet$_{(TMM2022)}$ [80] | ✔ | ✗ | ✔ | ✗ | 2.43/145.13 |
| CASNet$_{(TIP2022)}$ [5] | ✔ | ✗ | ✔ | ✔ | 16.90/128.97 |
| DPC-DUN$_{(TIP2023)}$ [60] | ✗ | ✗ | ✗ | ✔ | 11.44/93.38 |
| CAT-Net$_{(TMM2023)}$ [33] | ✗ | ✗ | ✗ | ✔ | 6.53/69.62 |
| OCTUF$_{(CVPR2023)}$ [61] | ✗ | ✗ | ✗ | ✔ | 3.74/102.57 |
| S$^2$-CSNet (Ours) | ✔ | ✔ | ✔ | ✔ | 11.59/108.37 |

[1] Adaptive CS Ratio Allocation (ACRA)   [2] Scale-Aware Adaptive Partitioning (SAP)
[3] Fine Granular Scalability (FGS)   [4] Without Access to Ground Truth Image (WAGTI)

**Table 4: The ablation results (PSNR) of different functional submodules of S$^2$-CSNet at different sampling rates on dataset Set11.**

| SAP | ACRA | DSAM | RD Map | Rate=0.01 | Rate=0.10 | Rate=0.25 | Avg. |
|---|---|---|---|---|---|---|---|
| ✗[1] | ✔ | ✔ | ✔ | 22.25 | 30.55 | 35.85 | 29.55 |
| ✗[2] | ✔ | ✔ | ✔ | 22.18 | 30.49 | 35.81 | 29.49 |
| ✔ | ✗ | ✔ | ✔ | 22.21 | 30.43 | 35.69 | 29.44 |
| ✔ | ✔ | ✗ | ✔ | 22.31 | 30.63 | 35.92 | 29.62 |
| ✔ | ✔ | ✔ | ✗ | 22.43 | 30.71 | 36.06 | 29.73 |
| ✔ | ✔ | ✔ | ✔ | **22.57** | **30.86** | **36.21** | **29.88** |

[1] $B = 16$   [2] $B = 32$

global features of the deep reconstruction network. In Table 4, we compare the native multi-scale network (i.e., composed of multiple sets of cascaded residual blocks) with our DSAM-based multi-scale network. It can be observed that DSAM can bring about 0.26dB gain compared to the non-DSAM strategy.

*4) Effect of RD Map $\mathcal{R}$:* In the process of deep reconstruction, we introduce the reconstruction distortion (RD) map (see Fig. 3) to perceive the distribution of distortion information. The RD map contains the reconstruction distortion of each block, which is used to guide the CS ratio allocation of S$^2$-CSNet. As shown in Table 4, the introduction of RD map results in an average PSNR gain of 0.15dB compared to inputting only the initial reconstructed image in the deep reconstruction.

## 5 Conclusion

In this paper, by analyzing the drawbacks of block-based sampling, a scale-aware scalable network (dubbed S$^2$-CSNet) for image compressive sensing is proposed, which achieves scale-variable adaptive sampling and fine granular scalability without direct access to the original image. Specifically, to adapt to multi-scale adaptive sampling, a structural sampling matrix with a predefined order is presented, which can sample multi-scale image blocks with arbitrary sampling rates. Then, we design a distortion-guided scale-aware scheme, which is used to evaluate the reconstruction distortion under different scale samplings and select the optimal scale for sampling. Furthermore, to reconstruct the measurements with high quality, a multi-scale hierarchical sub-network is developed, in which a dual spatial attention is embedded to mine the local and global affinities between dense feature representations. Extensive experiments on both objective metrics and subjective visual qualities demonstrate that the proposed S$^2$-CSNet outperforms existing state-of-the-art CS methods by large margins.

## Acknowledgments

This work was supported in part by the National Natural Science Foundation of China (NSFC) under grants 62076080, 62272128 and 62441202, in part by the Technique Program of Jiangsu under grant BE2021086.

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
