# OpenReview forum: "S$^2$-CSNet: Scale-Aware Scalable Sampling Network for Image Compressive Sensing"
_acmmm.org/ACMMM/2024/Conference — MM2024 Poster_

### Official Review · Reviewer_iBqk · 2024-05-01

**Rating:** 5
**Confidence:** 4

**Summary:**

This work introduces a novel approach named S2-CSNet for image compressive sensing (CS), addressing the two challenges of maximizing information preservation in linear CS measurements and accurately reconstructing original images. S2-CSNet utilizes a novel compressive sampling paradigm called scale-aware scalable sampling, where images are divided into non-overlapping blocks of sizes 16 and 32. These blocks are sampled using learnable universal sampling matrices in a self-adaptive, non-uniform manner while achieving the target sampling rate. A reconstruction distortion (RD) prior map, estimated from initial measurements, guides the adaptive CS ratio allocation in the subsequent sampling stage. The method employs a hierarchical grid network architecture that projects measurements back to the image domain and integrates them with the RD prior map for multi-scale reconstruction. Extensive experiments demonstrate that S2-CSNet achieves state-of-the-art performance across a wide range of sampling rates, validating the effectiveness of the proposed scale-aware adaptive sampling, dual spatial attention module (DSAM), and RD map.

**Strengths:**

1. The scale-aware adaptive sampling method introduces fine-grained scalability and is a novel approach in the community. It leverages different block sizes to enhance image information preservation within compressed measurements, significantly improving the efficiency and effectiveness of the sampling process. Currently, no existing algorithms offer similar insights into adaptive image content sampling.

2. S2-CSNet significantly outperforms existing methods, with gains of 0.4dB and 0.1dB PSNR on the Set11 and BSD68 datasets, respectively. These gains are achieved while maintaining comparable complexity in terms of parameter number and inference time.

3. The detailed analysis of the learned universal sampling matrices, as shown in Figures 7-10, provides valuable insights into the structural properties and distribution of matrix elements. These findings are not only interpretable but also beneficial for further research in the field.

**Limitations:**

Despite my overall positive assessment of this work, I recommend addressing the following concerns to strengthen the manuscript:

1. The manuscript lacks a detailed discussion on the practical implementation of scale-aware adaptive CS sampling in real-world scenarios, such as single-pixel cameras, digital image encoding, or encryption. The authors should provide a detailed example demonstrating how this framework could be applied in practical applications to ensure its relevance and utility.

2. The description of the scale-aware sampling process is complex and challenging to follow. I suggest adding a structured "Algorithm" part for the sampling process and relocating detailed network block designs to the supplementary material. This adjustment will help focus the main text on the primary contributions, improving readability and comprehension.

3. The method shows a more significant performance advantage at lower CS ratios. An explanation of why this occurs would provide deeper insights into the method's efficiency and limitations at different compression levels.

4. To aid in understanding and reproducing the results, I recommend providing access to the code and pre-trained models. This is crucial given the complexity of the proposed sampling and reconstruction methods.

5. The current manuscript omits several relevant and recent works that provide additional context and comparison for the proposed method. These include:

[1] Deep Physics-Guided Unrolling Generalization for Compressed Sensing

[2] Deep learning-enhanced snapshot hyperspectral confocal microscopy imaging system

[3] Deep Memory-Augmented Proximal Unrolling Network for Compressive Sensing

[4] SAUNet: Spatial-Attention Unfolding Network for Image Compressive Sensing

[5] ISTA-NET++: Flexible Deep Unfolding Network for Compressive Sensing

[6] D3c2-net: Dual-domain deep convolutional coding network for compressive sensing

**Suitability:**

3

---

### Official Review · Reviewer_xycJ · 2024-05-20

**Rating:** 3
**Confidence:** 3

**Summary:**

This paper presented a network for image compressive sensing, which mainly consists of three components. To be specific, a learnable matrix with a predefined order for mutli-scale sampling with arbitrary rates,  a distortion estimation method for guiding the sampling rate allocation, and a sampling scheme for completing multi-scale sampling.

**Strengths:**

1. In contrast to existing methods, the proposed method could achieve both variable sampling sizes and rates.
2. The proposed method achieved the moderate performance improvements compared to the baselines in the paper.

**Limitations:**

1. This work sets the number of scales to 2 (i.e., 16 and 32) by following the previous works. However, one of the advantages emphasized in the paper are handling multiple scales. The paper does not conduct experiments to verify the scalability of the method on 3 or more scales.
2. The proposed method handles multiple scales by respectively handling each of them, and selecting the scale with minimum reconstruction distortion. Such a way usually induces a poor scalability to model. How does the complexities of the model change with the number of block scales?
3. The differences and contrasts between this work and existing works were not clearly stated in the section of relate works. For example, what are the major differences between them in adaptive allocations of sampling rates? And why does the proposed strategy make sense and outperform the others?
4. Some typos exist in the paper, such as "reconstructiona".

**Suitability:**

2

---

### Official Review · Reviewer_1cBJ · 2024-05-24

**Rating:** 2
**Confidence:** 3

**Summary:**

This paper proposes a novel scale-aware CS network (dubbed S$^2$-CSNet), which achieves scale-aware adaptive sampling, fine granular scalability and high-quality reconstruction with one single model. However, the novelty is limited.

**Strengths:**

The paper indicates a completed story.

**Limitations:**

1)	Lns 38 and 39: What is the difference between "image compression" and "Image compressive sensing" in KEYWORDS?
2)	Notation used in the paper is very confused. Some expressions in this paper are ambiguous. Ln338: “Np is the number of samples”, Ln553: “Np is the number of training images”. Ln554: “  represents the reconstructed image  ”. Ln 189: “Q= lB2”,  Ln497: “the query feature map Q”, Ln487: “for the inputs Q and K”, Ln489: “To get the input tokens {˜Q,˜K,˜V}”. SA takes the query feature map Q as input, and the inputs Q indicates to query feature map or Token. It's hard to understand. What is the difference between the output of Eqs. (8) and (9)? Ln457: “e.g. they can calculate the outputs   and   of the DRBs, respectively.” This description is inconsistent with Eqs. (8) and (9). Figure 9: “first ten rows (1-10)”?
3)	Lns 383-386: “It can be seen that they are nearly linear relationship, i.e., the larger the ||Δx||, the larger the corresponding ||Δy|. Therefore, it is reasonable to infer ||Δx|| from ||Δy||.” As defined in Eq.4: “we propose a distortion estimation method in the measurement domain that only uses the sampling matrix to perfom linear operations on the measurements.” “linear operations” is predefined. Please provide a detailed theoretical analysis of "it is reasonable to infer ||Δx|| from ||Δy||".
4)	Ln 405: “Γ will select the scale Bs with the minimum distortion”. Please explain in detail the specific operation and selection strategy of Γ at scale Bs.
5)    References [70] and [71] are repeated.

**Suitability:**

2

---

### Official Review · Reviewer_m3Ni · 2024-05-26

**Rating:** 5
**Confidence:** 4

**Summary:**

This paper proposes a novel scale-aware scalable CS network, dubbed as $S^2$-CSNet, which achieves scale-aware adaptive sampling, fine granular scalability and high-quality reconstruction with a single model. The proposed $S^2$-CSNet makes use of a universal sampling matrix to achieve scalable sampling. It also makes use of a distortion-guided scale-aware scheme to select the optimal division scale for sampling. In $S^2$-CSNet, a multi-scale hierarchical sub-network is designed to reconstruct the image. The multi-scale hierarchical sub-network uses a dual spatial attention mechanism to mine the local and global affinities between dense feature representations. Extensive experiments on both objective metrics and subjective visual qualities demonstrate that the proposed $S^2$-CSNet outperforms existing state-of-the-art CS methods by large margins.

**Strengths:**

The proposed $S^2$-CSNet is the first deep network-based image CS that provides scale-aware adaptive sampling and fine granular scalability without direct access to the original image.

**Limitations:**

It is unclear how to generate RD map $\mathcal{R}$, which is an important input of deep reconstruction sub-module in the proposed $S^2$-CSNet.

According to this paper, the universal sampling matrix is a structural sampling matrix with a predefined order. What does the predefined order mean? How to decide this predefined order?

It is a little difficult to understand how the Scale-aware scalable sampling module works. The presentation of Sec 3.2 should be improved. Moreover, in Sec 3.2, Some formulas are incorrectly referenced, such as, at line 375-376, "the sampling of $S^2$-CSNet consists of two parts, the base sampling for Eq. (4) and the scale-aware sampling based on Eq. (4)."

**Suitability:**

3

---

### Meta-Review · Area_Chair_ZBNa · 2024-06-27

**Recommendation:** Accept (Poster)
**Confidence:** 4

**Metareview:**

1 Accept, 1 Weak Accept, 2 Borderline Reject.
All reviewers provide positive comments to this paper. However, several mineral issues are to be addressed in the final version.
-- how to generate RD map $\mathcal{R}$
-- What and how to determine the predefined order
-- how the Scale-aware scalable sampling module works
-- differences and contrasts between this work and existing works
-- detailed example demonstrating how this framework could be applied in practical applications
-- more experiments